# Response of *Bletilla striata* to Drought: Effects on Biochemical and Physiological Parameter Also with Electric Measurements

**DOI:** 10.3390/plants11172313

**Published:** 2022-09-04

**Authors:** Yongdao Gao, Chang Cai, Qiaoan Yang, Wenxuan Quan, Chaochan Li, Yanyou Wu

**Affiliations:** 1Key Laboratory for Information System of Mountainous Area and Protection of Ecological Environment of Guizhou Province, Guizhou Normal University, Guiyang 550001, China; 2State Key Laboratory of Environmental Geochemistry, Institute of Geochemistry, Chinese Academy of Sciences, Guiyang 550081, China

**Keywords:** *Bletilla striata*, drought stress, electrophysiological, morphological characteristics, cell metabolic energy

## Abstract

In heterogeneous landscapes with temporary water deficit characteristics in southwestern China, understanding the electrophysiological and morphological characteristics of *Bletilla striata* under different water conditions can help to better evaluate its suitability for planting plants in specific locations and guide planting and production. Using *B. striata* seedlings as experimental materials, the maximum field capacity (FC) was 75–80% (CK: control group), 50–60% FC (LS: light drought stress), 40–45% FC (MS: moderate drought stress), and 30–35% FC (SS: severe drought stress). In terms of physiological response, the activities of peroxidase (POD) and catalase (CAT) decreased under drought conditions, but the activity was well under the LS treatment, and the contents of proline (Pro) and malondialdehyde (MDA) increased. In terms of morphological responses, under drought conditions, root lengths of the rhizomes (except the LS treatment) were significantly reduced, the leaf lengths were reduced, and the biomass was significantly reduced. The stomatal size reached the maximum under the LS treatment, and the stomatal density gradually decreased with the increase in drought degree. In terms of electrophysiological responses, drought significantly decreased the net photosynthetic rate (*P*_N_) of *B. striata*, stomatal conductance (*g*_s_), and transpiration rate (*Tr*), but effectively increased the water use efficiency (*WUE*). The effective thickness of leaves of *B. striata* increased under drought conditions, and drought promoted the formation of leaf morphological diversity. Our results showed that drought stress changed the physiological and morphological characteristics of *B. striata*, and under light drought conditions had higher physiological activity, good morphological characteristics, higher cellular metabolic energy and ecological adaptability. Appropriate drought can promote the improvement of the quality of *B. striata*, and it can be widely planted in mildly arid areas.

## 1. Introduction

The degree and duration of drought affect the survival and development of plants [1,2,3], so the ability of plants to withstand water stress is crucial. With global climate change, drought has become an important factor for restricting global economic development and leads to declines in crop yields [4,5,6]. The response and adaptive characteristics of plants to the environment in the early stage of growth are important prerequisites for screening suitable pioneer plant populations and building a robust, stable and complex ecosystem. Ecological restoration species often choose species with strong drought resistance and growth ability, and water deficit prevents plants from maintaining normal growth, especially in the seedling stage of plants [7]. Therefore, those plants that use less water to grow are preferentially selected for land reclamation in arid areas. Drought reduces the morphological and physiological traits, reduces the leaf water potential [7] and saps movement due to alternation of xylem anatomical features in the plants, increases the content of osmotic substances in plants [8], affects the plant lactones [2], cell ion state, osmotic pressure [8,9], and reduces plant photosynthetic rate [5,10].

The mechanism of drought tolerance is very complex and when activated at different stages of plant development, it involves many physiological and biochemical processes at the cell, tissue, organ and whole plant level [6,11]. At the morphological level, drought has the greatest impact on stems and roots. Since roots are the only organ for plants to obtain water from soil, root length, leaf size and leaf biomass are common morphological traits related to drought resistance [12]. Under drought conditions, plants usually limit the number and area of leaves, thereby reducing biomass to maintain the water budget balance [13]. Biomass and root length are the most intuitive responses of plants to drought stress [3,14]. Deep and thick roots are conducive to water absorption [15,16], while small and thick leaves are more conducive to reducing leaf surface temperature [12,17]. In response to drought, plants evolved antioxidants and secondary metabolites to improve drought resistance [7]. For example, Pro plays an important role as an effective regulatory substance in plants, which can prevent cell dehydration and increase the concentration of cell fluid [18]. In addition, osmoregulation may provide seedlings with an ecological advantage to maintain plant metabolic activity under suboptimal conditions when roots have not yet reached deep soil moisture [19]. Drought stress reduces photosynthesis by reducing the unit leaf area and photosynthetic rate, which is mainly due to stomatal closure or metabolic disorders. Higher *WUE* is associated with stomatal conductance and stomatal opening and closing to reduce transpiration [16]. In this study, we investigated the medicinal value [20] and ornamental value of orchid *Bletilla striata* (Thunb. ex A. Murray) Rchb. f. (Figure 1), which is rich in polysaccharides and has the effects of hemostasis, swelling, and enhancement of body functions [21], is mainly used as a biomaterial in the medical field [22], and has a well-developed root system, and is often used for ecological restoration in the Karst area of Southwest China.

Electrophysiological indicators can represent the drought tolerance of plants and are widely used to detect the drought tolerance of plants [23]. Electrophysiology is an experimental method for electrical measurement and manipulation of living cells, living tissues and individuals. The earliest research on plant electrophysiology was made in 1873 by Sanderson, who discovered the transmission of electrical signals in the sensitive hairs of Dionaea Muscipula [24]. When plants are exposed to environmental or external stimuli, the electrical signals generated by plants are a direct response to environmental and external stimuli [25,26]. The plant defense strategy starts from the plasma membrane of plant cells. Biotic and abiotic interference induces chemical interactions of volatile organic compounds, triggering plant signal molecules, and imbalance of ion flux generated by plant plasma membrane in the sensing area. Such different charge distributions generate changes in the transmembrane potential [27]. When plants are subjected to water stress [28], the effects of these stressors on plants can be expressed by the plant electrical signals [29,30,31]. For example, leaf water potential [28], photosynthetic characteristics [23] and physiological capacitance (*C*) [24] of plants under drought stress were measured to quickly detect the growth state of plants under drought [5,28]. On the other hand, the energy required for plant growth and development is called the metabolizable energy of plant cells [32], which is the main form of energy to sustain life activities [33]. It reflects a series of assimilation and alienation processes in plants, including hydrogen exchange, assimilation and utilization of inorganic matter, synthesis and transformation of organic matter and energy, and all other physiological and biochemical processes in the body [34]. Plant action potential, physiological capacitance and physiological resistance (*R*) [24] and other indicators are important parameters for the study of plant electrophysiology. Using the Nernst equation, the chemical energy in cells can be expressed in the form of electrical energy. The changes of electrophysiological indicators reflect the changes in the metabolic energy of plant leaf cells to a certain extent, and can also reflect the ability of plants to resist stress under different stress [23,26,28]. Therefore, the electrophysiological characteristics of plants can be used to judge the adverse reactions of plants under adverse environmental conditions [25,35,36], which provides another method for the determination of the drought resistance of plants [37].

Karst habitats are highly heterogeneous, with high porosities of stratified limestone and low water retention capacities, which cause them to be highly susceptible to drought and pose a threat to plant growth [38]. In the context of decreasing global water resources, determining the optimal soil moisture content, and the corresponding status of *B. striata* can ensure precise irrigation amounts to maximize water conservation. Therefore, this study analyzed the response characteristics of *B. striata* seedlings to different drought treatments, established the correlation between the electrophysiological signals, plant physiology and morphological indicators of *B. striata*, and explored the adaptation strategies of *B. striata* seedlings to the drought environments. This research will help to more comprehensively understand their growth characteristics and drought tolerance under limited soil moisture conditions, thereby helping to establish a stable vegetation ecosystem in water-scarce environment in Karst areas and provide a reference for the selection of ecological restoration plants.

## 2. Results

### 2.1. B. striata Enzyme Activity Analysis

*B. striata* POD activity increased under the LS treatment and decreased under the SS treatment (Figure 2a). Compared with CK, the POD activity of B. striata was highest under the LS treatment (*p* < 0.01) and was significantly lower under the MS and SS treatments (*p* < 0.01).

Figure 2b shows that the CAT activity of *B. striata* decreased (*p* < 0.01) with increasing drought. Compared with the CK, the LS treatment had the highest activity of *B. striata* CAT (*p* < 0.01), and the CAT activity was lowest under the SS treatment (*p* < 0.01), which showed that drought stress decreased the CAT activity.

Drought stress promoted the production of *B. striata* Pro (*p* < 0.01). Figure 2c shows that the *B. striata* Pro contents exhibited a gradual increase with increasing drought, with the highest Pro contents occurring in the SS treatment group and the lowest Pro contents occurring in the CK.

Figure 2d shows that the *B. striata* MDA contents gradually increased (*p* < 0.05). Compared with the CK, the LS treatment had a lowest MDA content, but the difference relative to the CK was nonsignificant (*p* > 0.05). The SS treatment had a highest MDA content, and the differences between the CK, LS and MS treatments were significant (*p* < 0.05).

### 2.2. B. striata Root Lengths and Biomass Analysis

Drought stress significantly inhibited the elongation of *B. striata* root lengths. Compared with the CK, the root lengths of *B. striata* were elongated under the LS treatment, but the difference was nonsignificant (*p* > 0.05), while the root lengths became shorter under the MS (*p* < 0.05) and SS treatments (*p* < 0.01), which were significantly different from the CK (Figure 3a). Drought stress significantly inhibited *B. striata* leaf lengths (*p* < 0.05). Compared with the CK, the lengths of *B. striata* leaves became shorter in the LS, MS and SS treatments, but with an intensification in drought conditions, the lengths of leaves exhibited an elongation trend, and the overall performance was in the order of SS > MS > LS (Figure 3a).

Drought stress significantly reduced the biomass of the roots and leaves of *B. striata* (*p* < 0.01). Compared to the CK, the biomass of *B. striata* significantly decreased under the LS, MS and SS treatments (*p* < 0.05) (Figure 3b).

### 2.3. Analysis of B. striata Photosynthetic Characteristics

Drought affected the photosynthetic characteristics of *B. striata* to different degrees (Figure 4). Drought stress significantly affected the *P*_N_ of *B. striata* (*p* < 0.01). The *P*_N_ under the LS, MS, and SS treatments were significantly lower than that under the CK, with the highest rate occurring under the SS treatment and the lowest rate occurring under the MS treatment (Figure 4a). Drought stress significantly reduced the *g*_s_ of *B. striata* (*p* < 0.05), while the *g*_s_ was highest for the LS treatment, but was not significantly different from that of the CK and was lowest for the SS treatment, which was significantly different from the CK (Figure 4b). Drought stress significantly reduced the intercellular CO_2_ concentrations (*C_i_*) in *B. striata* (*p* < 0.01); the *C_i_* in the LS, MS and SS treatment were lower than that in the CK, while the lowest value was for SS treatment, which differed significantly from the CK, LS, and MS treatments, and indicated that drought stress began to suppress the *C_i_* (Figure 4c). The *B. striata Tr* gradually decreased with increasing drought gradients (*p* < 0.05); the *B. striata Tr* under the LS, MS and SS treatments were significantly lower than that for the CK, and this result indicated that drought stress caused it to reduce *Tr* to retain leaf water (Figure 4d).

Drought stress significantly reduced the light use efficiency (*LUE)* of *B. striata* (*p* < 0.01). The *LUE* under the LS, MS and SS treatment were significantly lower than that of the CK; the highest *LUE* was observed for the SS treatment and the lowest was observed for the MS treatment, which was significantly different from that of the CK (Figure 4e). Drought stress significantly affected the *B. striata WUE* (*p* < 0.05) and when compared to the CK, both of the *WUE* under the drought adversity treatments were higher than that under the CK, while the highest *WUE* occurred for the SS treatment (Figure 4f).

### 2.4. B. striata Physiological Capacitance Model

As the clamping force F increases, the *C* of the *B. striata* leaves increases, and the *C* changes linearly with the clamping force (Figure 5).

Origin 2018 was used to dynamically fit the experimental data, obtain the parameters of $\;x_{0}\;$ and *h* in the equation $C=x_{0}+hF$, obtain the function parameters and equations of the *C* of the leaves and the clamping force F, and obtain the statistical data of the equation fit. The values of *R*^2^, *n* and *p* are shown in Table 1.

### 2.5. B. striata Physiological Resistance Model

As the clamping force F increases, the *R* of the *B. striata* leaves decrease, and the *R* decreases exponentially with the change of the clamping force F relationship (Figure 6).

Using Origin 2018 to dynamically fit the experimental data, the parameters of *y*_0_, *k*_1_ and *b*_1_ in the equation *R* = *y*_0_ + $e^{-b_{1}F}$ are obtained, and the *R* of the *B. striata* leaves are obtained as a function of the clamping force F parameters and equations; at the same time, the equation-fitting statistics *R*^2^, *n* and *p* values are obtained (Table 2).

### 2.6. B. striata Physiological Impedance (Z) Model

As the clamping force F increases, the *Z* of the leaf decreases, and the *Z* decreases exponentially with the change of the clamping force F (Figure 7).

Origin 2018 was used to dynamically fit the experimental data, and obtain the parameters of *p*_0_, *k*_2_ and *b*_2_ in the equation *Z* = *p*_0_
+ ${\;e}^{-b_{2}F}$, and obtain the *Z* of *B. striata* leaves as a function of the clamping force F parameters and equations, and simultaneously obtain the *R*^2^, *n* and *p* values that the equations fit the statistical data (Table 3).

### 2.7. Cell Metabolic Energy Analysis

Compared with CK, the *d* of LS, MS and SS treatment increased (Table 4). Δ*G*_R-E_, Δ*G*_Z-E_, Δ*G*_R_, Δ*G*_Z_ and Δ*G*_B_ showed a trend of first increasing and then decreasing with the aggravation of drought. Among them, the Δ*G*_R-E_, Δ*G*_Z-E_, Δ*G*_R_, Δ*G*_Z_ and Δ*G*_B_ of LS were larger than CK, MS and SS treatments (Table 4).

### 2.8. B. striata Stomatal Feature Analysis

The *B. striata* stomatal density of the CK was significantly higher than those of the LS, MS and SS (Figure 8 and Figure 9), while the stomatal lengths of the CK were smaller than those of the LS and SS, but the stomatal widths were significantly higher than those of the remaining treatments. Figure 9(a_1_,a_2_) shows that the veins on the back of the *B. striata* leaves in the CK were clear, with a dorsal texture distribution of the leaf veins, and the back pattern was more regular and pentagonal, while the stomata were distributed inversely according to the leaf veins. The leaf surfaces showed no obvious tomenta, the keratin layer was smooth, the stomata were not sunken, with a large air hole opening and closing, the stomata were surrounded by rough and uninterrupted annular or pinnate striate projections, and the stomatal apparatus projected from the epidermis and exhibited an oval shape.

The veins in the lower abaxial part of the leaves of the *B. striata* LS treatment were clear, the dorsal texture was distributed by the leaf veins, the back pattern was irregular and pentagonal, and the leaf surfaces had no obvious tomenta. The distribution of the *B. striata* stomata was more regular in the LS, and the density of the leaf stomata decreased in the LS treatment compared to the CK group (*p* < 0.05) (Figure 8 and Figure 9), partial stomatal closure occurred, and the stomata became narrower and longer but were shorter in width (Figure 8 and Figure 9). Wrinkling of the epidermal part of the leaf was observed in addition to a smooth and flat inner source of the arch cover inside the air hole, and the exogenous stratum corneum bulged into ridged thickening and unsmoothed into feathery interrupted enclosing stomata (Figure 9(b_1_,b_2_)).

Compared with the CK, the *B. striata* stomatal density of the MS treatment was significantly lower (*p* < 0.01), partial closure of the stomata was present, the stomatal lengths and widths were significantly shorter, the stomata were sunken with smooth and flat inner sources of the arch covers inside the air holes, the exogenous cuticle was thickened and raised into ridges, which were unevenly interrupted by plumes that surrounded the stomata, and there were increased numbers of pentagonal grain folds (Figure 9(c_1_,c_2_)).

In the SS treatment, the pattern on the back of the *B. striata* leaves was irregularly diamond-shaped or no longer had a specific shape, and the surface texture of the leaves gradually disappeared and crumpled significantly. The stomata near the leaf veins all showed certain degrees of shrinkage and disappearance, the stomata became narrower and longer, the stomatal widths were significantly lower than those of the CK, LS and MS treatment, and the stomatal sagging and closure were significantly higher in the SS treatment. The stomatal densities were significantly lower than that of the CK, but were higher than those of the LS and MS treatment; the stomata were closed, the inner arch cover was smooth and flat on the inner source, and the outer cuticle was thickened and raised into a ridge and was uneven into a feathery interruption that surrounded the stomata (Figure 9(d_1_,d_2_)).

## 3. Discussion

### 3.1. Response of B. striata Enzyme Activities to Drought Stress

Drought stress affects photosynthesis in plants [5,10,16], and this disturbance can lead to oxidative stress responses and the accumulation of reactive oxygen species (ROS) in plant cells. ROS is ubiquitous in cells, and its production is limited to chloroplasts, mitochondria and peroxisomes [39]. One of the initial responses of plants to water deficit is the formation of ROS [40,41], expressed as membrane peroxidation [42]; among them, H_2_O_2_ is an important signalling molecule in ROS and plays a role as a signalling molecule in the process of plant stress [43]. Plants produce ROS when exposed to the stress conditions, such as drought stress [44], which forced the plants to produce antioxidants [39,41,44], flavonoids, and secondary metabolites which play a role in protecting the plant by detoxifying ROS and protecting the plant from abnormal conditions (i.e., stress) and protein and amino acid stabilization [7,45]. POD and CAT are important protective enzymes in plants that can effectively scavenge H_2_O_2_ (^1^O_2_), (•OH) and (O_2_^−^) radicals to prevent membrane lipid peroxidation when plants are stressed by the external environment [45,46]. Under drought stress, the increase in Pro in leaves of *B. striata* was significant (Figure 2c), which is a general response of plants to abiotic stress. Pro is a compatible solute that acts as a penetrant and hydroxyl radical scavenger to protect cells from stress damage [39,43]. In response to drought, oxidative damage (MDA) increased in the leaves of *B. striata*, and decreased under mild drought conditions (Figure 2d). To some extent, the production of MDA can be used to indicate the resilience of plants [47,48].

### 3.2. Morphological Characteristics of B. striata under Drought Environments

The water deficit treatment of *B. striata* revealed decreases in root biomass and leaf biomass when compared to the CK group (Figure 2), which was similar to a previous study on *Vatairea macrocarpa* (Benth.) [49]. In response to drought stress, plants usually limit the number and areas of leaves, which thus reduces the biomass and water balance at the expense of yield loss [13]. At the morphological level, drought has the greatest effects on stems and roots, and since roots are the only organs of plants that obtain water from the soil, the biomass quantities and root lengths are the most intuitive responses of plants to drought stress [3], and deep and thick root systems facilitate water uptake [15,16]. Unlike the results of previous studies [15,16], no elongation trends in root lengths were observed in this study with the deepening of drought, which may be related to the growth morphology of *B. striata*. *B. striata* roots are stem tubers, and when subjected to drought stress, the plant water is mainly stored in the stem tubers. To maintain the normal physiological functions, such as leaf photosynthesis, the water stored in the stem tubers is exported to the aboveground part of the leaves, so any root length growth is not obvious. However, these roots can withstand prolonged drought and produce new functional root systems after rehydration.

### 3.3. Response of B. striata Photosynthetic Characteristics to Drought Stress

Drought stress affects the ultrastructure of leaves, especially the characteristics of stomata. Stomatal control is the main physiological mechanism of plant drought resistance, which protects plants from irreversible damage caused by hydraulic failure and carbon starvation by regulating photosynthesis and transpiration [7,50], and is closely related to the physiological activities of plants such as photosynthesis, transpiration and respiration [16]. Stomatal closure is widely considered to be the main determinant of photosynthesis reduction under drought conditions [51,52]. Stomatal density and size are related to transpiration unit leaf area, but the contribution of stomatal morphology varies with environmental conditions, and smaller stomatal size may be beneficial under some conditions [53]. The reasons for stomatal opening, closing, and partial opening may alter transpiration rates at the leaf canopy level. Stomatal width was significantly correlated with *Tr* (Figure 10), and the decrease in stomatal width in *B. striata* may also be the main cause of *P*_N_ inhibition and gas exchange limitation [16,54]. In this study, drought stress significantly inhibited *B. striata*’s stomatal length, stomatal width and stomatal density (Figure 8a and Figure 9), however, under SS treatment, the stomatal density was abnormally higher than that of LS and MS treatment. In the case of smaller stomatal morphology, *B. striata* benefits from changes in stomatal morphology by increasing the number of stomata to obtain more CO_2_ [16], while reducing transpiration.

Many internal and external factors can lead to *g*_s_ adjustment, which Cowan and Farquhar have previously proposed to ensure that CO_2_ intake is much greater than evaporation [53,55]. This critical process at the plant–environment interface becomes critical when soil moisture conditions are limited. In the present study, the decrease in *g*_s_ was one of the main factors for the decrease in *P*_N_. Correlation analysis showed that *B. striata g*_s_ was positively correlated with *C_i_* (Figure 10), and under limited *C_i_* conditions, the photosynthetic carbon metabolism changed, and light reactions led to the accumulation of photosynthetic electron transport components, a decrease in molecular oxygen, an increase in ROS, and ROS caused *B. striata* photosynthetic organ damage [16] and a decrease in the photosynthetic rate.

A higher *WUE* was associated with *g*_s_ and stomatal opening to reduce transpiration [16,55], and the correlation analysis showed that the *g*_s_ values were highly positively correlated with the *Tr* (Figure 10). In *B. striata,* a decrease in *g*_s_ was associated with a decrease in the *Tr* and also with a decrease in water loss. The first choice of plants when coping with drought is to close the stomata; when the stomata are closed, little CO_2_ is absorbed, and transpiration is reduced [55]. In the present study, the *B. striata Tr* decreased when compared to those of the CK (Figure 4d), while the *g*_s_ and *WUE* values decreased, which was similar to the results of a previous study on *Hibiscus rosa-sinensis* under drought stress [56]. The water-deficient treatment of *B. striata* exhibited a higher *WUE* (Figure 4f), which indicated that *B. striata*, when facing drought conditions, adjusts the amount of water loss by opening and closing the stomata and sacrificing CO_2_ absorption, which thereby increasing the instantaneous *WUE*, which also explains the higher *WUE* (Figure 4f) and the larger numbers of closed stomata (Figure 9(d_1_,d_2_)) under severe drought conditions.

### 3.4. Response of B. striata Electrophysiology to Drought Stress

The amount of water in plants affects the elasticity vulnerability of plant leaf cells. When plants are subjected to different degrees of environmental stress, the water status of cells will change immediately, and the electrical signal of plants will also change [23]. In general, the greater the effective thickness of the plant, the larger the vesicles, the more mature the leaves, and the stronger the water storage capacity [57]. In the present study, the effective thickness of the leaves of *B. striata* increased with the increase in the drought degree, and the main reason may be that large vesicles were formed to store more water to cope with drought. Therefore, electrophysiological indicators can reflect the leaf water status of plants [23,26]. The electrical change in plant photosynthesis is an electrical reaction caused by metabolic changes, showing differences in oxygen production and carbon dioxide consumption, and potential differences between different parts. Therefore, changes in electrophysiological indicators can reflect changes in photosynthesis and metabolism of plants. In this study, the characteristics of electrophysiological parameters changed significantly. With the increase in drought, the physiological capacitance of *B. striata* gradually increased (Figure 5), *P_N_* decreased, transpiration decreased, and *WUE* increased. The changes in photosynthetic indices reflected the unique relationship between physiological impedance and photosynthetic index of *B. striata*; the greater the soil water deficit, the greater the physiological capacitance, and the lower the *g*_s_, *Tr* and *LUE* of plants. Under drought conditions, the reduction in water availability usually results in the limited uptake of total nutrients by crops, thereby reducing the tissue concentration of plants [16]. The physiological capacitance and physiological resistance [26,28] are significant differences, so there are differences in the metabolic energy of plant cells. The process of material and energy distribution during plant growth and development and the adaptive mechanism of plants under stress are closely related to the source–sink relationship and plant cell metabolizable energy is another expression of plant physiological activity. From Table 4, it can be seen that the highest cellular metabolic energy of *B. striata* leaves under LS treatment, leaf cells maintain sufficiently high energy reserves to eliminate ROS, allowing the plant to adjust its metabolism and generate an appropriate adaptive response [44]. Therefore, *B. striata* under light drought conditions has a more flexible source–sink relationship, and has a strong adaptability to drought conditions. The change characteristics of its cellular metabolizable energy may be an effective stress marker under environmental constraints. Such changes in physiology, photosynthetic properties, cellular metabolic energy, and stomatal characteristics of *B. striata* allow them to better adapt to a wider range of environmental stresses.

## 4. Materials and Methods

### 4.1. Experimental Materials and Experimental Design

The experiments were conducted at the Guizhou Normal University campus (106°43′08″ E, 26°35′31″ N), with an altitude of 1096 m, average annual temperature of 15.0 °C, annual sunshine duration of 1252.8 h and average annual relative humidity of 82%; the soil used for the experiment consisted of yellow loam. *B. striata* tissue culture seedlings were purchased from Guizhou Anlong Biotechnology Co., Ltd. (Guizhou, China). The pot planting method was adopted (pot height 12 cm, pot bottom diameter 8 cm, and pot mouth diameter 15 cm); the annual *B. striata* seedlings with consistent growth were selected as the test materials and were randomly divided into the following four groups, the maximum field capacity (FC) was 75–80%, 50–60% FC, 40–45% FC, 30–35% FC, which were recorded as control group (CK), light drought stress (LS), moderate drought stress (MS) and severe drought stress (SS); each group consisted of 5 pots with 1 plant per pot; each pot was weighed to maintain the relative soil moisture content within the defined range [52]. This process took 3 weeks. The soil water contents were tested every day to replenish any deficient amounts.

### 4.2. POD Activity Assay

POD, CAT, Pro, and MDA were measured using UV-Visible spectrophotometers (UV-Vis Spectrophotometer 6010, Agilent Technologies Shanghai Co.). Measurements of POD activity was performed according Thongsook [58] and Mou [59], with appropriate adjustments. Catalyzed by peroxidase, H_2_O_2_ oxidizes guaiacol to a brown product, which has a maximum absorbance at 470 nm. *B. striata* leaves (0.5 g) were ground into a homogenate in an ice mortar with phosphate buffer, transferred the homogenate into a centrifuge tube and centrifuged at 8000 r/min for 15 min at 4 °C. The supernatant was transferred to a volumetric flask, the volume fixed at 25 mL and stored at low temperature. The enzyme solution boiled in hot water was used as a control, and the enzyme reaction system consisted of 3 mL of 0.05 mol/L pH = 7.0 phosphate buffer, 1.0 mL of 0.3% H_2_O_2_, 1.0 mL of 0.2% guaiacol and 0.1 mL of enzyme solution. The absorbance was measured at 470 nm, with A470 in each minute. A change of 0.01 was considered as 1 unit of peroxidase activity.

### 4.3. CAT Activity Assay

H_2_O_2_ has a strong absorption at the wavelength of 240 nm, and the activity of catalase can be measured by measuring the rate of change of absorbance. First, 0.5 g of leaves were added to 0.1 mol/L pH = 7.0 phosphate buffer and ground into a homogenate, centrifuged to take the supernatant, 0.2 mL enzyme solution was added, 1.5 mL pH = 7.8 phosphoric acid, 1 mL distilled water, 0.3 mL 0.1 mol/L H_2_O_2_, the absorbance was recorded at 240 nm. One unit of CAT activity was defined as the amount of enzyme that used 1 μmol H_2_O_2_ min^−1^ [60].

### 4.4. Pro Content Assay

Pro was determined by the colorimetric method using hydrated ninhydrin. When plant samples were extracted with nitro salicylic acid, Pro was free in the salicylic acid solution, and the solution turned red after heating with acidic ninhydrin. Then the pigment was transferred with toluene, colorimetry was performed at 520 nm, and then the amount of Pro was checked from the standard curve [61].

### 4.5. MDA Content Assay

MDA content was determined by the thiobarbituric acid (TBA) colorimetric method [55]. *B. striata* leaf samples (0.5 g) were added to 5% trichloroacetic acid (TCA) to grind into a homogenate, the homogenate was centrifuged at 8000 r/min for 20 min, and the supernatant was the sample extract. Next, 1 mL of supernatant was placed in a test tube, 2 mL of 0.67% TBA was added, and the mixture was mixed well and boiled for 30 min in a 100 °C water bath. The liquid was centrifuged at 10,000 r/min, and the supernatant was transferred to a cuvette, subtract the absorbance at 600 nm from the absorbance at 532 nm [60,62,63].

### 4.6. B. striata Biomass and Growth Parameters Assay

The roots length and leaves length of *B. striata* were measured by tape measure, accurate to 0.01; the fresh weight of *B. striata* roots and fresh weight of leaves were measured by analytical balance, accurate to 0.01.

### 4.7. B. striata Photosynthetic Parameters Assay

Using the American portable photosynthetic system Li-6400XT (LI-COR Inc., Lincoln, NE, USA), three pots of each plant of CK, LS, MS, and SS groups of *B. striata* were selected for the determination of photosynthetic parameters in October 2019. The *P*_N_, *g*_s_, *C_i_*, *Tr*, *WUE* and *LUE* of the *B. striata* leaves were recorded during the measurements of the photosynthetic parameters in sunny and well-lit daytime periods at 10:00–11:00 am. The average atmospheric temperature during the measurement period was 21.19 °C, the average atmospheric CO_2_ concentration was 283.95 μmol·mol^−1^, and the average atmospheric relative humidity was 32.43%.

### 4.8. Establishment of Electrophysiological Model and Acquisition of Cell Metabolic Energy

Using an LCR tester (model 3532-50, HIOKI, Nagano, Japan), referring to Wu [57], the physiological capacitance, physiological resistance, and physiological impedance of the *B. striata* leaves were measured. The effect of the LCR tester on plants The best measurement frequency is 3 KHz, and the best measurement voltage is 1.5 V. The specific effective thickness *d* (10^−12^ m) of *B. striata* leaf was calculated based on *C*, *R*, *Z*, the unit metabolic energy of plant leaf cell based on *R* Δ*G*_R-E_ (J/m), and the unit metabolism of plant leaf cell based on *Z* energy Δ*G*_Z-E_ (J/m), plant leaf cell metabolic energy Δ*G*_R_ (10^−12^ J) based on *R*, plant leaf cell metabolic energy Δ*G*_Z_ (10^−12^ J) based on *R*, and plant leaf cell metabolic energy Δ*G*_B_ (10^−12^ J).

### 4.9. Physiological Capacitance

The cytosol in the plant has a cellular vesicle solute, and the vesicle solute in the plant will change accordingly when it is stressed by adversity, while the cellular solute in the plant leaves is used as a dielectric, and the conductivity of the vesicle solute can be well judged according to the plant adversity [26,37,57]. The *C* of plant leaves is obtained by increasing the clamping force by stacking iron blocks, and the change in clamping force leads to a change in cytosolic solute concentration in leaves and a change in cytosolic solute conductivity of *B. striata* leaf tissue [26,57].

By the gravimetric equation:(1)F=(M+m)g
where F is the clamping force, N; M is the mass of the iron block, m is the mass of the plastic rod and electrode sheet, kg; and g is the acceleration of gravity of 9.8, N/kg.

The Gibbs free energy equation is expressed as ΔG=ΔH+PV, the energy equation of a capacitor $W=\frac{1}{2}U^{2}C$, and the equation of the plant cell subjected to pressure $P=\frac{F}{S}$.

W is the energy of the capacitor, which is equal to the work transformed by the Gibbs free energy Δ*G*, W = Δ*G*; ΔH is the internal energy of the leaf system, P is the pressure on the leaf cells of *B. striata*, V is the volume of the leaf cells of *B. striata*, U is the voltage parameter of the test setup, and *C* is the physiological capacitance of *B. striata*. Where the pressure equation F is the clamping force and S is the effective area under the action of the pole plate, so the *C* of the plant leaves varies with the clamping force F model as
(2)$$C=\frac{2\Delta H}{U^{2}}+\frac{2V}{{\mathrm{SU}}^{2}}F$$

Let $x_{0}=\frac{2\Delta H}{U^{2}}$, $h=\frac{2d}{U^{2}}$, and Equation (2) can be deformed as
(3)$$C=x_{0}+\mathrm{Hf}$$

### 4.10. Physiological Resistance

When external drought stress occurs, plant leaf water decreases, affecting changes in cytosol water thus leading to changes in ion concentration inside and outside the plant leaf, affecting the transmission of resistive current ions. The relationship between the physiological resistance of plant cells and drought conditions in leaves was derived from the Nernst equation based on the difference between internal and external ion concentrations [57].

The expression of the Nernst equation is given by
(4)$$E-E^{0}=\frac{R_{0}T}{n_{R}F_{0}\;}\ln\frac{C_{i}}{C_{o}}$$

The internal energy of the electric potential E can be converted into work performed by pressure, which is proportional to PV, PV = a E, i.e.,
(5)$$\mathrm{PV}=\mathrm{Ae}={a\;E}^{0}+\frac{{a\;R}_{0}T}{n_{R}F_{0}\;}\ln\frac{C_{i}}{C_{o}}$$
where P is the pressure on the plant cell, a is the electric potential conversion energy coefficient, V is the plant cell volume; and P is the pressure on the plant body cell, $P=\frac{F}{S}$, where F is the clamping force and S is the effective area under the action of the pole plate.

Thus, (5) can become
(6)$$\frac{V}{S}F={a\;E}^{0}-\frac{{a\;R}_{0}T}{n_{R}F_{0\;}}\ln\frac{C_{T}R-f_{0}}{f_{0}}$$

Further deformation, we can obtain:(7)$$R=\frac{f_{0}}{C_{T}}+\frac{f_{0}}{C_{T}}e^{\frac{n_{R}F_{0}E^{0}}{R_{0}T}}e^{\left(-\frac{{\mathrm{Vn}}_{R}F_{0}}{{\mathrm{Sa}\;R}_{0}T}F\right)}$$

In Equation (8), R is the physiological resistance, and since d $=\frac{V}{S}$, Equation (8) can be deformed as follows:(8)$$R=\frac{f_{0}}{C_{T}}+\frac{f_{0}}{C_{T}}e^{\frac{n_{R}F_{0}E^{0}}{R_{0}T}}e^{\left(-\frac{{d\;n}_{R}F_{0}}{{a\;R}_{0}T}F\right)}$$

In Equation (8), d, a, E^0^, R_0_, T, n_R_, F_0_, C_T_, and f_0_ are fixed values, let $y_{0}$ = $\frac{f_{0}}{C_{T}}$, k_1_ = $\frac{f_{0}}{C_{T}}e^{\frac{n_{R}F_{0}E^{0}}{R_{0}T}}$, b_1_ = $\frac{{d\;n}_{R}F_{0}}{{a\;R}_{0}T}$, thus, the Equation (8) can be deformed as
(9)$$R=y_{0}+k_{1}\;e^{-b_{1}F}$$


In Equation (9), y_0_, k_1_ and b_1_ are the parameters of the model. Therefore, the metabolic energy per unit of plant leaf cell based on physiological resistance Δ*G*_R-E_ = $\frac{{a\;E}^{0}}{d}=\frac{\ln k_{1}-\ln y_{0}}{b_{1}}$. Metabolic energy of plant leaf cell based on physiological resistance Δ*G*_R_ = Δ*G*_R-E_d.

### 4.11. Physiological Impedance

The magnitude of impedance in the same plant depends on the concentration of cytosolic solute in response to the physiological impedance electrolyte, and when the plant is subjected to external environmental changes, the ensuing changes in cellular electrolytes lead to changes in physiological impedance inside and outside the membrane, obeying the Nernst equation [57].

Expression of the Nernst equation.
(10)$$E-E^{0}=\frac{R_{0}T}{n_{Z}F_{0}\;}\ln\frac{Q_{i}}{Q_{o}}$$

The internal energy of the electric potential E can be converted into work performed by pressure, which is proportional to PV, PV = a E, i.e.,
(11)$$\mathrm{PV}=\mathrm{aE}={a\;E}^{0}+\frac{a\;R_{0}T}{n_{Z}F_{0}\;}\ln\frac{Q_{i}}{Q_{o}}$$

Where P is the pressure on the plant cell, a is the electric potential conversion energy coefficient, V is the plant cell volume; the pressure P on the plant cell can be found by the pressure formula, where F is the clamping force and S is the effective area under the action of the pole plate.

Therefore, (11) can become
(12)$$\frac{V}{S}F={a\;E}^{0}-\frac{{a\;R}_{0}T}{n_{Z}F_{0\;}}\ln\frac{\mathrm{QZ}-J_{0}}{J_{0}}$$

Further deformation, we can obtain:(13)$$Z=\frac{J_{0}}{Q}+\frac{J_{0}}{Q}e^{\frac{n_{Z}F_{0}E^{0}}{R_{0}T}}e^{\left(-\frac{{\mathrm{Vn}}_{Z}F_{0}}{{\mathrm{Sa}\;R}_{0}T}F\right)}$$

In Formula (14), Z is the physiological impedance. Since d $=\frac{V}{S}$, Formula (13) can be transformed into:(14)$$+\frac{J_{0}}{Q}e^{\frac{n_{Z}F_{0}E^{0}}{R_{0}T}}e^{\left(-\frac{{d\;n}_{Z}F_{0}}{{a\;R}_{0}T}F\right)}$$
where d, a, E^0^, R_0_, T, n_Z_, F_0_, Q, J_0_ are all fixed values, let $p_{0}$ = $\frac{J_{0}}{Q}$, k_2_ = $\frac{J_{0}}{Q}e^{\frac{n_{Z}F_{0}E^{0}}{R_{0}T}}$, b_2_ = $\frac{{d\;n}_{Z}F_{0}}{{a\;R}_{0}T}$, thus Equation (14) can be deformed as
(15)$$Z=p_{0}+k_{2}\;e^{-b_{2}F}$$
where p_0_, k_2_ and b_2_ are the parameters of the model. Therefore, the metabolic energy per unit of plant leaf cell based on physiological impedance Δ*G*_Z-E_ = $\frac{{a\;E}^{0}}{d}=\frac{\ln k_{2}-\ln p_{0}}{b_{2}}$. Metabolic energy of plant leaf cells based on physiological impedance Δ*G*_Z_ = Δ*G*_Z-E_d. Δ*G*_B_ is the average of Δ*G*_R_ and Δ*G*_Z_.

### 4.12. B. striata Stomatal Scanning

The drought-treated *B. striata* leaves were removed, the upper and lower epidermal dust on the leaves was wiped off with skimmed cotton, a blade was used 3/1 from the main leaf veins to take sample sizes of approximately 3 mm × 3 mm size from the leaves, double-sided adhesive was fixed in the sample table, and a JFC-1600 sputtering instrument was sprayed with platinum. The samples were placed on a scanning electron microscope (KYKY-1000B, Shimadzu, Kyoto, Japan), and the cellular structure characteristics of the *B. striata* leaves were observed at ×1000 magnification. The *B. striata* leaf structures were photographed by a digital microscope imaging system and were calibrated by the digital ranging software, Motic Images Advanced 3.0. The stomatal density was measured by taking digital photos of visual fields in different directions at ×100 magnification. The number of stomata in each visual field was counted (four visual fields were averaged), and the area of the visual field was obtained, so stomatal density (mm^2^) = stomata number/S. Where visual field area: The length and width of the scanning image of the electron microscope are 22.8 cm and 16.5 cm, and the length of the 100 μm ruler is 1.7 cm. The actual length of the field of view is 22.8/1.7 × 100 μm = 1341.18 μm, and the actual width of the field of view is 16.5/1.7 × 100 μm = 970.59 μm. Visual field area =1341.18 μm × 970.59 μm = 1,301,735.90 μm^2^ = 1.30 mm^2^.

### 4.13. Statistical Analysis

Data were processed with MS-Excel^®^ 2016 (Microsoft Inc., Redmond, WA, USA) and SPSS 23 (SPSS Inc., Chicago, IL, USA) software. One-way analysis of variance (ANOVA) and the least significant difference test (LSD) were used for significance testing (*p* < 0.05). The data are shown as the means ± SE. Graphs were prepared using Origin 2018 (Northampton, MA, USA). The R program was used for plotting, and the program packages used were “ggplot2” and “GGally”, The detailed code is as follows:

install.packages(“ggplot2”)

install.packages(“GGally”)

library(ggplot2)

library(GGally)

ppo<-read.csv(choose.files(),header = TRUE,row.names = 1)

ggpairs(iris,columns=1:4,aes(color=Species),upper=list(continuous=wrap(‘cor’,size=3)),lower = list(continuous=‘smooth’))iris

install.packages(“corrplot”)

library(corrplot)

ppo<-read.csv(choose.files(),header = TRUE,row.names = 1)

M<-cor(ppo)

res1<-cor.mtest(M,conf.level=.95)

res2<-cor.mtest(M,conf.level=.99)

p.mat=res1$p

corrplot(M,order=‘hclust’,type=‘upper’,tl.pos=‘d’,p.mat=res1$p,sig.level=c(0.001,0.01,0.05),insig = ‘label_sig’,pch.cex=1,tl.srt = 0,tl.cex = 0.5,cl.cex = 1)

corrplot(M,add=TRUE,type=‘lower’,method=‘number’,order=‘hclust’,diag=FALSE,tl.pos=‘n’,number.cex = 0.58)

## 5. Conclusions

For Karst areas with seasonal water shortages, the reasonable selection of cultivated species is beneficial to regional plant restoration and economic cost savings. Under drought stress conditions, the physiological characteristics and morphology of *B. striata* changed significantly, showing low *P*_N_, *g*_s_, *Tr*, *LUE* and high *WUE*. In *B. striata*, the morphological adaptation mode was small and thick leaves and long roots, and the water utilization strategy was to increase stomatal length, decrease stomatal density and increase cell metabolized energy under light drought conditions. *B. striata* under mild drought showed more obvious morphological and physiological avoidance and adaptation to arid environments. It is generally believed that plants grow well in an environment adapted to soil moisture. However, in this study, severe drought and a humid environment were not conducive to the growth of *B. striata*, and proper drying (i.e., LS treatment) can promote the improvement of its quality. There was a significant correlation between the electrophysiological indexes and photosynthetic indexes of *B. striata*, which could jointly characterize the response of the growth and development of *B. striata* to soil water, but the correlation between the electrophysiological indexes and photosynthetic indexes was not obvious. In the general trend, the growth index and electrophysiological index of *B. striata* were consistent. Therefore, plant electrophysiological information can be used as another means to judge the state of plant soil water deficit. These findings will be beneficial to the selection of cultivated species in arid areas and the rapid monitoring of plant drought.

## Figures and Tables

**Figure 1 plants-11-02313-f001:**
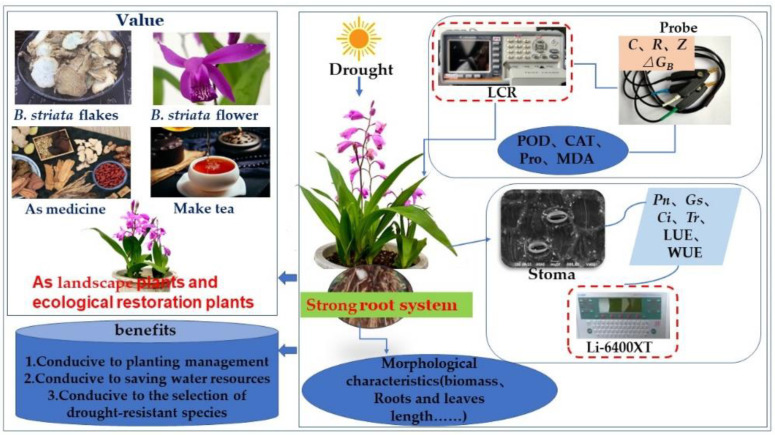
*B. striata* application value and research ideas.

**Figure 2 plants-11-02313-f002:**
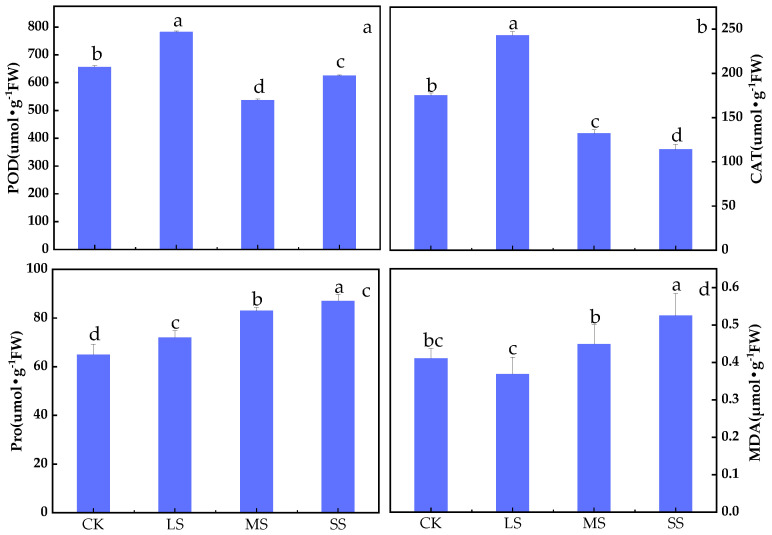
Effects of drought stress on POD, CAT, Pro, and MDA in *B. striata*. Peroxidase (**a**), catalase (**b**), proline (**c**), malondialdehyde (**d**). CK = control group, LS = light drought stress, MS = moderate drought stress, and SS = severe drought stress. The bars indicate the standard errors of the means (± SE). The different lowercase letters indicate significant differences according to LSD test.

**Figure 3 plants-11-02313-f003:**
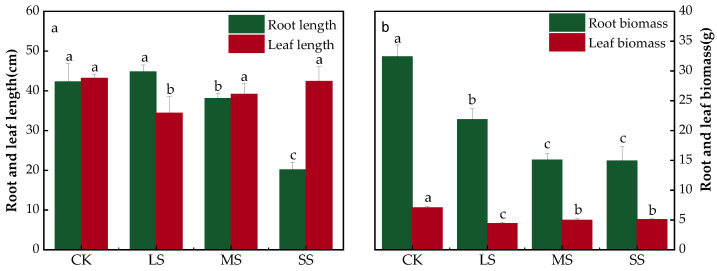
Effects of drought stress on root length (**a**), leaf length and biomass (**b**). The bars indicate the standard errors of the means (±SE). The different lowercase letters indicate significant differences according to LSD test.

**Figure 4 plants-11-02313-f004:**
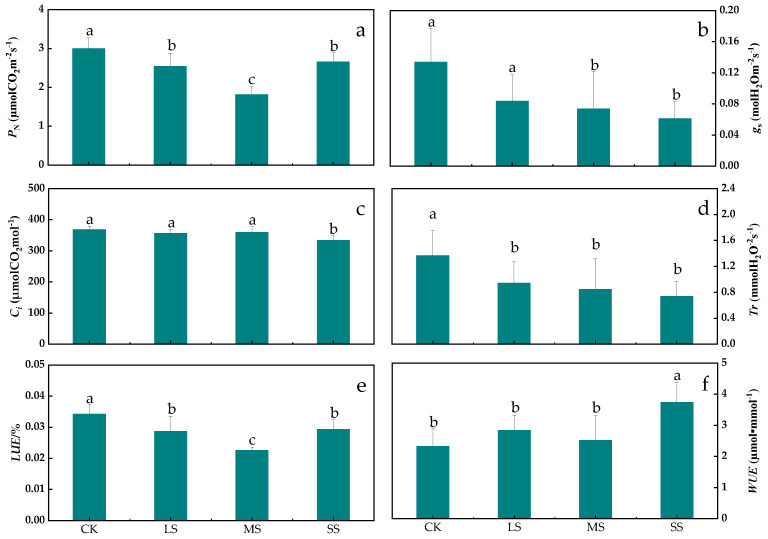
Responses of the photosynthetic parameters of *B. striata* leaves to drought stress. The bars indicate the standard errors of the means (± SE). The different lowercase letters indicate significant differences according to LSD test. *P*_N_: Net photosynthetic rate (**a**); *g*_s_: Stomatal conductance (**b**); *C_i_*: Intercellular CO_2_ concentration (**c**); *Tr*: Transpiration rate (**d**); *LUE*: Light use efficiency (**e**); *WUE*: Water use efficiency (**f**).

**Figure 5 plants-11-02313-f005:**
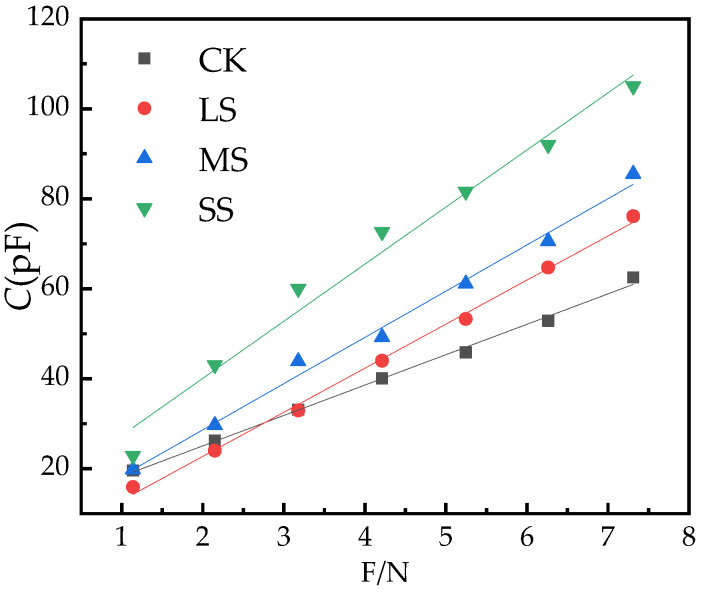
Variation of *C* of *B. striata* leaf with clamping force F.

**Figure 6 plants-11-02313-f006:**
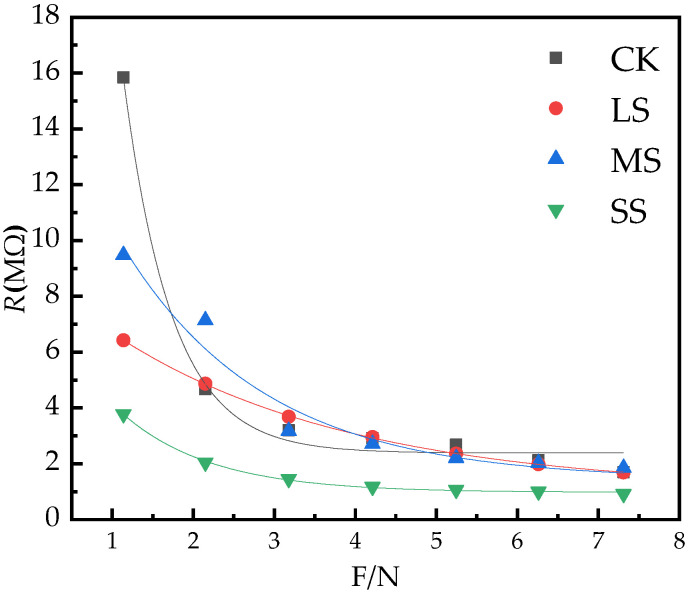
The change of *B. striata* leaf *R* with clamping force F.

**Figure 7 plants-11-02313-f007:**
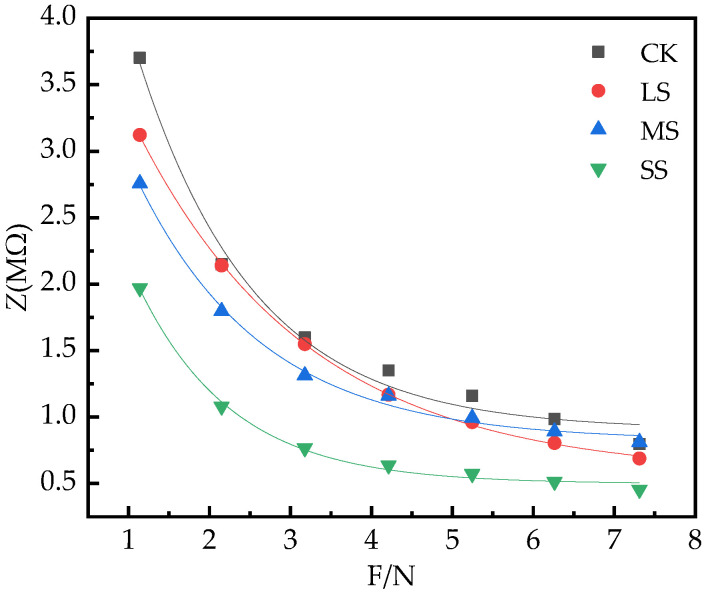
The change of *B. striata* leaf *Z* with clamping force F.

**Figure 8 plants-11-02313-f008:**
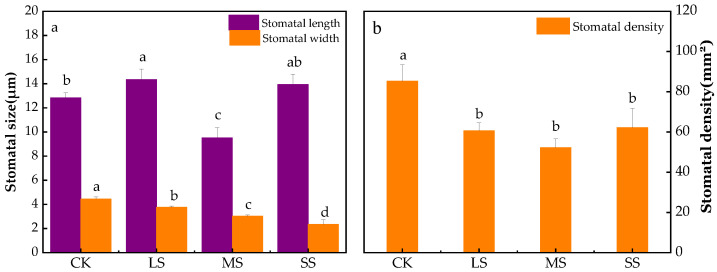
Effect of drought stress on the *B. striata* stomatal. Stomatal length and stomatal width (**a**), stomatal density (**b**). The bars indicate the standard errors of the means (±SE). The different lowercase letters indicate significant differences according to LSD test.

**Figure 9 plants-11-02313-f009:**
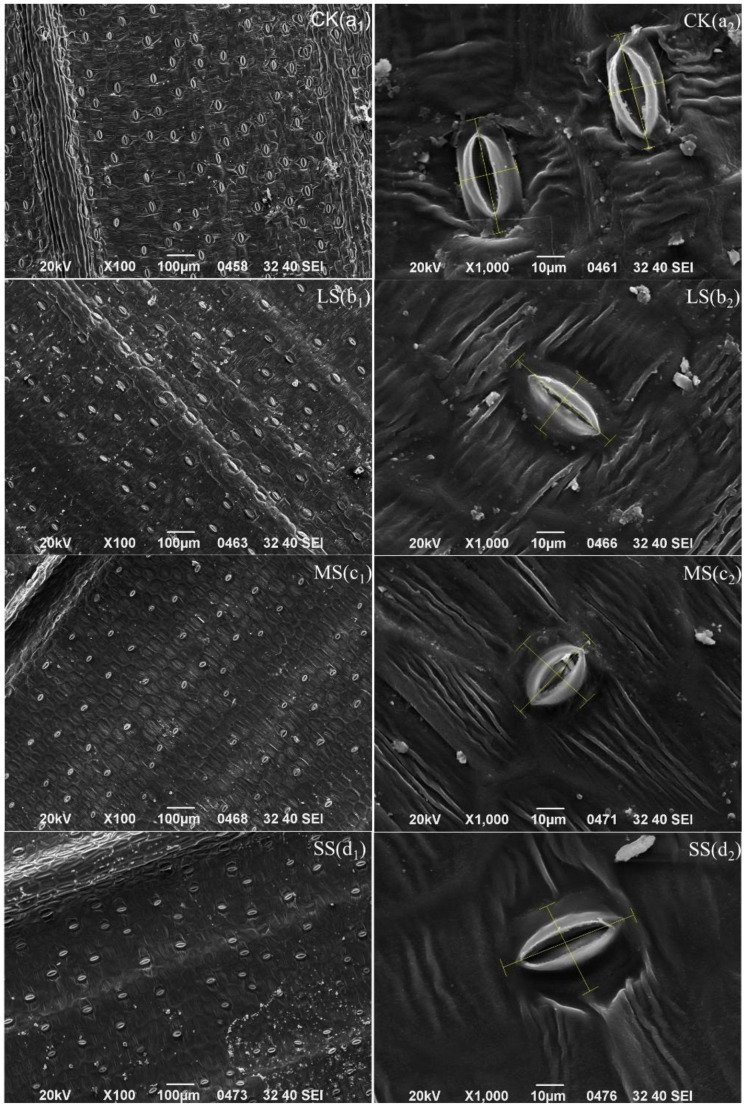
Electron microscope scans of *B. striata* stomatal pores. Stomata density (**a_1_**,**b_1_**,**c_1_**,**d_1_**); stomata size (**a_2_**,**b_2_**,**c_2_**,**d_2_**).

**Figure 10 plants-11-02313-f010:**
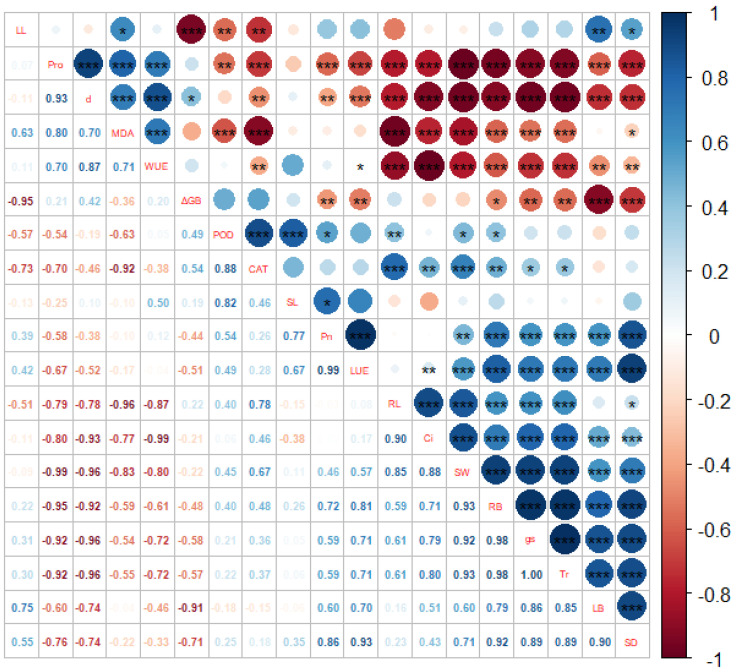
Correlation analysis of various of *B. striata* indicators. The R program was used for plotting, and the program packages used were “ggplot2” and “GGally”. The blue colors in the graph indicate that the correlations are positive, the red colors indicate that the correlations are negative, and “*” indicates the significance level.

**Table 1 plants-11-02313-t001:** The equations and parameters of the model of the leaves *C* with the clamping force (F).

Treatments	*x* _0_	*h*	Equation
CK	11.54	6.70	*C* = 11.54 + 6.70F *R*^2^ = 0.99, *p* < 0.05, *n* = 76
LS	3.12	9.80	*C* = 3.12 + 9.80F *R*^2^ = 0.99, *p* < 0.05, *n* = 76
MS	8.06	10.28	*C* = 8.06 + 10.28F *R*^2^ = 0.99, *p* < 0.05, *n* = 76
SS	14.68	12.70	*C* = 14.68 + 12.70F *R*^2^ = 0.98, *p* < 0.05, *n* = 76

**Table 2 plants-11-02313-t002:** The equations and parameters of the model of the leaves *R* with the clamping force (F).

Treatments	*y* _0_	*k* _1_	*b* _1_	Equation
CK	2.39	90.82	1.68	*R* = 2.39 + 90.82e^−1.68F^ *R*^2^ = 0.99, *p* < 0.05, *n* = 76
LS	1.05	7.94	0.34	*R* = 1.05 + 7.94e^−0.34F^ *R*^2^ = 0.99, *p* < 0.05, *n* = 76
MS	1.42	15.94	0.57	*R* = 1.42 + 15.94e^−0.57F^ *R*^2^ = 0.95, *p* < 0.05, *n* = 76
SS	0.97	7.82	0.90	*R* = 0.97 + 7.82e^−0.90F^ *R*^2^ = 0.99, *p* < 0.05, *n* = 76

**Table 3 plants-11-02313-t003:** The equations and parameters of the model of the leaves *Z* with the clamping force (F).

Treatments	*p* _0_	*k* _2_	*b* _2_	Equation
CK	0.91	6.06	0.69	*Z* = 0.91 + 6.06e^−0.69F^ *R*^2^ = 0.98, *p* < 0.05, *n* = 76
LS	0.57	4.34	0.47	*Z* = 0.57 + 4.34e^−0.47F^ *R*^2^ = 0.99, *p* < 0.05, *n* = 76
MS	0.82	3.99	0.64	*Z* = 0.82 + 3.99e^−0.64F^ *R*^2^ = 0.99, *p* < 0.05, *n* = 76
SS	0.50	3.89	0.86	*Z* = 0.50 + 3.89e^−0.86F^ *R*^2^ = 0.99, *p* < 0.05, *n* = 76

**Table 4 plants-11-02313-t004:** *d* and cell metabolic energy of *B. striata* leaves under different treatments.

Treatments	*d*/(10^−12^m)	Δ*G*_R-E_/(10^12^Jm^−1^)	Δ*G*_Z-E_/(10^12^Jm^−1^)	Δ*G*_R_/(J)	Δ*G*_Z_/(J)	Δ*G*_B_/(J)
CK	7.54	2.17	2.75	16.32	20.71	18.52
LS	11.03	5.95	4.32	65.60	47.62	56.61
MS	11.57	4.24	2.47	49.06	28.59	38.83
SS	14.29	2.32	2.39	33.13	34.08	33.61

## Data Availability

Data are available from the authors upon request.

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
