# Peer review of "Response of Bletilla striata to Drought: Effects on Biochemical and Physiological Parameter Also with Electric Measurements"

_plants, 2022, doi:10.3390/plants11172313_

Round 1

Reviewer 1 Report

General comments

I have read the manuscript (plants-1841531). Entitle: Characteristics of electrophysiological response of Bletilla striata to drought environments written by Yongdao Gao et. al., for publication of plants MDPI. In this study, the author investigates the enzyme activity, biomass, photosynthetic characteristics, leaf cell metabolic energy and leaf stomata morphology to analyze the response of B. striata to drought stress. In this study author mainly found that they inhibited the peroxidase (POD) and catalase (CAT) activities, while drought significantly increased the contents of proline (Pro) and malondialdehyde (MDA). Drought stress inhibited root biomass more than leaves. Additionally, the author also found that the photosynthesis of B. striata is inhibited under drought stress and net photosynthetic rate (PN), stomatal conductance (gs), transpiration rate (Tr) decreased, and stomatal density, but the water use efficiency (WUE) effectively increased.

The overall research is well conducted, and research is obvious application potential for the readers because of B. striata in a light drought environment indicating its high physiological activity and ecological adaptability in a light drought environment, which can be used as a species for Karst ecological restoration. In this sense, the manuscript is much valuable. However, I found some points, especially the flow of the text and lack of potential references, and lack of connection of story in different paragraphs, especially in the introduction and discussion sections. The author should provide enough examples and their interpretation of different traits of physiological and biochemical responses by the latest and appropriate references, some of which I mentioned below. Overall after I evaluate and request the author for this manuscript as a “MAJOR REVISION”.

 Major suggestions

 1)  Introduction: The introduction is well starting with the background of climate change and change in earth water cycle which is much appreciated. However, the author should mention the overall effect of drought, especially in shrubs or seedlings in the introduction, or the most suited initial paragraph. Please read and mentioned this as a reference. Entitle: Entitle “Response of drought stress in prunus sargentii and larix kaempferii ...https://doi.org/10.1016/j.foreco.2020.118099” Please mentioned that “drought reduced the morphological and physiological traits, reduce the leaf water potential and sap movement due to alternation of xylem anatomical features in the plants”. Then only write the background of Bletilla striata of medically important and other benefits.

2) Hypothesis and objectives of the study: Author should rephrase the text more deeply in the last paragraph of the introduction specially Ln. 59 to 63 by focusing on the hypothesis of the study and also connecting the objectives of the study. Author tries to mention but this is still not clear. The hypothesis of the study is an important thing, and it gives another strength to the introduction. The hypothesis should be very clear in the introduction sections because, without appropriate literature, questions, or hypotheses in the introduction section the entire text will be unclear. The author should give special attention and the sequential presentation of the content in the introduction with presenting the hypothesis of the study.

3) Discussion (Line no. 209 to 226): Author should Improve the subsection of discussion 3.1 more logically with clearer potential references because the main theme of antioxidant and secondary metabolites under drought stress conditions and release the ROS (why ROS is emerging in stress conditions?). Refer to these two articles for better clarify (1) https://doi.org/10.1038/s41598-019-55889 (2) https://doi.org/10.1016/j.scitotenv.2021.146466 and mention somewhere in that paragraph “abiotic stress especially environmental stress (I.e. drought) plant produces the ROS when these plant exposed to the stress condition and plant produce antioxidant, flavonoids, and secondary metabolites play to the role for protecting the plant for detoxifying ROS and protect the plant to protect the abnormal condition (i.e. stress) and protein and amino acid stabilization”.

 Some other comments

4) Line no. 455: Author should make more clearance of the text related to the counting of the stomata (stomatal density). How many replications and how much area is used by the author for the stomatal density and how author further process the final result of it?

5) Line no. 244: Generally, the greater the stomatal features such as higher length, the wider ts width, and other related features. The stromal opening, closing, and partial opening cause may also vary in the transpiration rate from the leaf canopy level. I recommend this article https://doi.org/10.1016/j.envexpbot.2020.104111  for the citation o describe the story of stomal opening and closing and also mention the role of stomatal morphological features and their contribution to Pn and related traits in the discussion section because the author did not focus on this part well in the discussion but this is very important highly required to mention in the discussion section.

6) Line no. 467: The author should to detail mention the statistical software and its detailed data analysis parts. Such what level of significance, software detail. Moreover, author used the R programming for some of the figures in this manuscript but I do not see its detail in the statistical section.

7) Line no. 470 (Conclusion): The conclusion for me comes off as repetitive of the abstract or a summary of the results section. I would love to read striking points and take-home messages that will linger in the readers’ minds. What is the novelty, how does the study elucidate some questions in this field, and the contributions the paper may offer to the scientific community?

8) Line no. 495 (Reference): please double-check the citations, their style, spell check, and other grammatical errors. moreover, I request to the authors for revision throughout the manuscript according to the journal rules.

Good Luck!

Author Response

请参阅附件。

Reviewer 2 Report

My comments are below.

 Abstract 

The abstract is carelessly written authors should incorporate their notable findings and adequately connect with the sentences they choose to correspond.

Introduction

  • The introduction section must have a clear hypothesis and significantly develop the second paragraph of your manuscript. Make it more connecting to the problem statement. 
  • Overall there is the repetition of the information, which could be avoided.

Discussion 

  • This section should include more information and references related to the relevant and related works. 

Conclusions

  • If possible, restructure and carefully edit the conclusion section and add clear information regarding the most noteworthy findings.

Round 2

Reviewer 1 Report

Dear Author

I have read the revised manuscript (plants-1841531). Titled: Characteristics of electrophysiological response of Bletilla striata to drought environment for publication of plant MDPI. This is the second submission made by the author. The author addressed all the questions and suggestions that I raised the issue in the review of the original manuscript. I satisfy the author’s revisions throughout the paper. Author well addresses the abstract issues. Especially author improved the introduction and discussion section very well inflow. Now, this manuscript improved the flow of writing, which was comparatively shallow in the original version but in this revised copy author addressed all the quarries and suggestions very well. Before accepting this manuscript if there is anything needed to be revised by the author, especially English grammar, or spell check, I request this manuscript is currently in “Minor Revision” and any grammatical error author may improve in this stage. Thank you.

Author Response

Dear Reviewers,

We are delighted and honored to receive your comments again and thank you for giving us the opportunity to make "Minor revisions" to the manuscript. We would like to thank the reviewers for spending a lot of time to read our revised articles and recognize our revisions. Your constructive comments have greatly improved the quality of the manuscript, which is the greatest encouragement to our research work and the driving force for our continuous progress. I fully accept your review comments, based on these comments and suggestions, we downloaded the latest version of the manuscript from the journal server, we conducted a grammar and spelling check on the full text, and invited a professional professor to conduct a second grammar and spelling check, and the revised manuscript has been resubmitted to your journal. All changes made to the text are shown in red. For details, please refer to the tracking change manuscript (plants-1841531-Track Changes 2) uploaded to the journal.